# Balanced Actor Initialization: Stable RLHF Training of Distillation Reasoning Models

## Abstract

The development of alignment and reasoning capabilities in large language models has seen remarkable progress through two paradigms: instruction tuning and reinforcement learning from human feedback (RLHF) alignment paradigm, and distillation-based reasoning fine-tuning paradigm. While both approaches prove effective independently, the third paradigm of applying RLHF to distillation-trained models presents significant challenges. Our investigation reveals two critical phenomena that emerge in this paradigm: Sequence Length Collapse, where language generation dramatically reduces during early RLHF training, and the Reward Hockey Stick Curve, featuring severe reward score drops followed by gradual recovery. These instabilities fundamentally compromise the model's alignment and reasoning capabilities. To address these challenges, we propose Balanced Actor Initialization (BAI), a two-stage weighted model merging approach. BAI first merges instruction-following and distillation-based reasoning fine-tuned models, then further combines this intermediate model with the pretrained model to preserve foundational knowledge. Through comprehensive experiments across diverse benchmarks and detailed analysis of training experiments, we demonstrate that BAI resolves Sequence Length Collapse, mitigates the Reward Hockey Stick Curve, and enables continuous sequence length improvement during training. Our analysis reveals that balanced merging ratios achieve optimal trade-offs between training stability and reasoning capability preservation. Our work provides the effective solution for stable training in this third paradigm, enabling more capable reasoning models that combine distillation efficiency with RLHF alignment.

## 1 Introduction

The development of alignment and reasoning capabilities in Large Language Models (LLMs) has emerged as one of the most critical challenges in modern artificial intelligence (OpenAI, 2024; Xu et al., 2025; Sui et al., 2025; Liu et al., 2025). Recent breakthroughs in chain-of-thought (CoT) reasoning (Wei et al., 2022; Chen et al., 2025) have demonstrated the potential for models to engage in step-by-step problem solving, leading to improvements across diverse reasoning tasks. Moreover, the recent success of DeepSeek-R1 (Guo et al., 2025) has demonstrated remarkable capabilities in reasoning and problem-solving, showcasing the potential of advanced post-training methodologies.

Current approaches to developing alignment and reasoning capabilities in language models typically follow two well-established paradigms. As shown in Figure 1, **Paradigm 1** is the instruction tuning and alignment paradigm, which involves supervised fine-tuning on instruction-following data followed by reinforcement learning from human feedback (RLHF) to align model behavior with human preferences (Ouyang et al., 2022; Bai et al., 2022a; Touvron et al., 2023; Achiam et al., 2023; Team et al., 2023). **Paradigm 2** is the distillation-based reasoning fine-tuning paradigm, where models are trained on reasoning data distilled from more powerful models (Guo et al., 2025). This approach enables smaller models that originally lack thinking and reasoning capabilities to acquire sophisticated step-by-step reasoning abilities through supervised learning on distilled data. This paradigm has proven highly effective because distilling reasoning capabilities from larger models into smaller ones yields excellent results with significantly lower computational costs compared to training smaller models through large-scale reinforcement learning (Luo et al., 2025; Li et al., 2025a). Following the breakthrough of DeepSeek-R1, this paradigm has become increasingly prevalent, with recent works

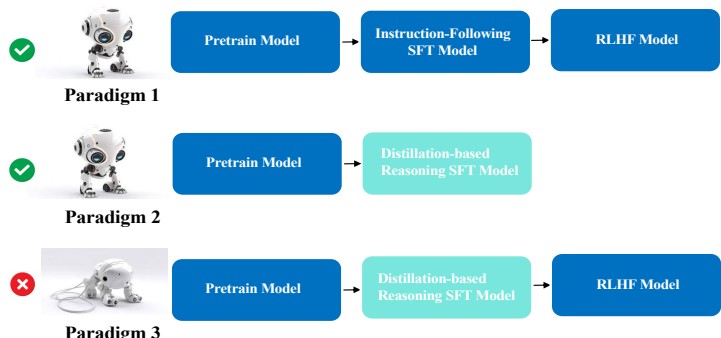

Figure 1: The LLMs training pipeline of Paradigm 1 (instruction tuning and RLHF alignment), Paradigm 2 (distillation-based reasoning fine-tuning), and Paradigm 3 (distillation-based reasoning fine-tuning and RLHF alignment).

demonstrating substantial performance improvements across various reasoning benchmarks through the use of large quantities of long chain-of-thought data distilled from giant-sized language models.

Given the success of both paradigms independently, a natural question emerges: can we achieve further breakthroughs by combining the instruction tuning and alignment paradigm with distillation-based reasoning fine-tuning? **Paradigm 3**—applying RLHF to models that have already undergone distillation-based reasoning fine-tuning-represents a potentially powerful methodology for developing reasoning models that combine the efficiency of distillation with the alignment benefits of human feedback optimization. However, this third paradigm presents significant challenges. Our experiments reveal that applying RLHF to models trained with extensive distillation-based reasoning fine-tuning leads to critical training instabilities.

Specifically, models initially generate lengthy reasoning chains after distillation-based reasoning fine-tuning, but after the first several steps of Proximal Policy Optimization (PPO) based reinforcement learning training, the response length experiences a dramatic reduction. We term this phenomenon **Sequence Length Collapse**. Simultaneously, we observe what we call the **Reward Hockey Stick Curve**, where reward model scores dramatically drop during early RL training before gradually recovering. These phenomena stem from the fundamental mismatch between specialized reasoning patterns learned during distillation-based fine-tuning and RL optimization requirements, often triggering reward hacking behaviors where models exploit reward signals through shortcuts rather than genuine reasoning improvement. This degradation fundamentally compromises the model's ability to produce detailed reasoning chains and comprehensive responses, representing a critical barrier to successful implementation of the third paradigm.

Guo et al. (2025) has shown that incorporating a small amount of cold-start data before reasoning-oriented RL training significantly improves training stability, highlighting the critical importance of robust model initialization. The significance of this cold-start data lies in creating a more balanced model initialization that preserves the model's foundational capabilities while introducing basic reasoning patterns, thereby establishing a stable foundation for subsequent reinforcement learning. Motivated by this observation, we recognize that robust actor model initialization is essential for addressing the instability issues in the third paradigm. To create such robust initializations and address the Sequence Length Collapse and Reward Hockey Stick Curve phenomena, we propose an effective weighted model merging approach, which we call Balanced Actor Initialization (BAI), that creates robust actor model initializations by combining the pretrained model with instruction-following fine-tuned models and reasoning fine-tuned models at different ratios. This weight merging approach provides a more deterministic and controllable initialization scheme, eliminating the dependence on ambiguous data quantity specifications while offering precise control over the balance between foundational capabilities and reasoning skills.

Specifically, our proposed BAI approach includes two stages to create robust actor initializations. In the first stage, we merge the instruction-following SFT model and the distillation-based reasoning SFT model through weighted linear combination to integrate both instruction-following capabilities and reasoning abilities. In the second stage, we further combine this intermediate model from the

first stage with the pretrained model to preserve foundational knowledge while maintaining the acquired specialized abilities. This two-stage approach directly addresses the challenges of integrating specialized reasoning abilities while preventing the degradation of foundational model capabilities.

Our comprehensive experiments demonstrate that BAI approach successfully resolves Sequence Length Collapse and effectively mitigates the Reward Hockey Stick Curve phenomenon, while providing better control and interpretability. Moreover, by addressing these core instabilities, our simple but effective approach enables the third paradigm to deliver improved performance across diverse evaluation domains. Through extensive BAI ratio experiments, we demonstrate that different merging configurations achieve distinct trade-offs between training stability and alignment and reasoning capability, with balanced ratios demonstrating optimal performance across diverse tasks.

Our paper makes the following key contributions:

- We identify Sequence Length Collapse and Reward Hockey Stick phenomena that emerge in the third paradigm, providing empirical analysis of their impact on training stability.

- We propose Balanced Actor Initialization (BAI), a two-stage weighted merging approach that addresses these instability issues, enabling continuous sequence length improvement while maintaining model performance.

- We demonstrate through extensive experiments that BAI achieves stable training with improved sequence length maintenance, gradual reward increases, and enhanced knowledge retention across diverse benchmarks.

## 2 RELATED WORK

The standard RLHF pipeline consists of three stages: SFT on instruction-following, reward model training using human preference comparisons, and policy optimization using reinforcement learning algorithms. This approach has proven highly effective for improving model helpfulness, harmlessness, and honesty (Bai et al., 2022b), leading to the success of models like ChatGPT (Achiam et al., 2023), Claude (Bai et al., 2022a), and Gemini (Team et al., 2023).

Recent advances in RLHF have focused on developing more effective and stable optimization algorithms. Proximal Policy Optimization (PPO) (Schulman et al., 2017) remains the most widely used approach, providing stable policy updates through clipped objective functions. Direct Preference Optimization (DPO) (Rafailov et al., 2024) eliminates the need for explicit reward model training by directly optimizing preferences, simplifying the pipeline while maintaining competitive performance. Group Relative Policy Optimization (GRPO) (Shao et al., 2024) improves sample efficiency by leveraging group-wise preference comparisons. DAPO (Yu et al., 2025) provides an open-source RLHF system designed for large-scale deployment, while VAPO (Yue et al., 2025) focuses on efficient and reliable reinforcement learning specifically for advanced reasoning tasks.

The emergence of reasoning-capable models has introduced new challenges and opportunities in RLHF. Recent thinking models demonstrate remarkable capabilities in step-by-step reasoning through explicit chain-of-thought generation. OpenAI's o1-style reasoning models (OpenAI, 2024) pioneered this direction by incorporating sophisticated reasoning protocols into the RLHF framework. Following this breakthrough, DeepSeek-R1 (DeepSeek-AI et al., 2024) demonstrated the successful application of RLHF to reasoning models, showing significant improvements in mathematical and logical reasoning tasks. Subsequently, models like SEED-1.5-Thinking (Seed et al., 2025) have further advanced the field by developing superb reasoning capabilities through reinforcement learning approaches.

However, training thinking models presents unique challenges, particularly regarding initialization strategies and training stability. Unlike traditional language models, reasoning models require careful balance between maintaining reasoning capabilities learned during SFT and adapting to human preferences through RL (Zheng et al., 2024a;b). DeepSeek-R1 (DeepSeek-AI et al., 2024) observed that training directly from base models leads to unstable cold start phases, while using a small amount of long CoT data for initialization improves stability. Unlike previous approaches that focus on data-based solutions, we propose BAI as a more controllable approach for achieving stable RLHF training while preserving reasoning capabilities.

## 3 APPROACH

As shown in Figure 1, Paradigm 1 represents the traditional instruction tuning and alignment approach, Paradigm 2 employs distillation-based reasoning fine-tuning without subsequent RLHF, and Paradigm 3 combines distillation-based reasoning fine-tuning with RLHF. However, Paradigm 3 faces critical training instabilities that compromise model performance. These instabilities manifest in two ways: models experience dramatic sequence length reduction during early RL training, losing their ability to generate detailed reasoning chains, while simultaneously exhibiting severe reward model score fluctuations that disrupt the learning process.

To address the Sequence Length Collapse and Reward Hockey Stick Curve, we propose Balanced Actor Initialization (BAI), a two-stage weighted model merging approach that creates robust initializations for RL training. BAI combines multiple models to achieve an optimal balance between reasoning capabilities, instruction-following abilities, and foundational knowledge retention.

### 3.1 BALANCED ACTOR INITIALIZATION (BAI)

Our approach focuses on creating robust actor model initializations through strategic model merging before RL training begins. The core motivation of BAI is to address the initialization challenges inherent in the third paradigm by leveraging the complementary strengths of different model states while mitigating their individual limitations. BAI operates in two distinct stages, each addressing different aspects of the initialization challenge. The first stage focuses on capability integration, combining specialized fine-tuned models to create a unified representation of reasoning and instruction-following abilities. Notably, this stage is flexible and can accommodate scenarios where only the distillation-based reasoning model is available, without requiring additional instruction-following components. The second stage emphasizes knowledge preservation, merging the integrated model with the original pretrained model to retain foundational capabilities that are crucial for stable RL optimization. This hierarchical design ensures that the final initialization maintains the delicate balance required for successful training in the third paradigm.

### 3.1.1 MULTI-SFT MODEL MERGING

First, given $N$ well-trained SFT models with different capabilities for merging, we denote the parameters of the $i$-th model as $\mathbf{M}_i^{\text{sft}}$ for $i \in \{1, 2, \ldots, N\}$. Each well-trained model is assigned a weighting coefficient $w_i$ that determines its contribution to the final merged model. The merged SFT model $\mathbf{M}_{\text{merge}}^{\text{sft}}$ is then computed as a weighted linear combination:

$$\mathbf{M}_{\text{merge}}^{\text{sft}} = \sum_{i=1}^{N} w_i \mathbf{M}_i^{\text{sft}} \tag{1}$$

where the weights sum to one $\sum_{i=1}^{N} w_i = 1$ to preserve parameter scale. This formulation allows for flexible integration of instruction-following and reasoning capabilities from different fine-tuning stages, enabling optimal balance between specialized capabilities while maintaining model stability. In this work, our first stage, Multi-SFT Model Merging, effectively combines the instruction-following SFT model from Paradigm 1 with the distillation-based reasoning SFT model from Paradigm 2 using uniform weights ($w_1 = 0.5$, $w_2 = 0.5$) to retain both strong instruction-following and reasoning capabilities.

### 3.1.2 BALANCED MODEL MERGING FOR RL ACTOR INITIALIZATION

While the merged SFT model $\mathbf{M}_{\text{merge}}^{\text{sft}}$ possesses strong instruction-following capabilities, it often suffers from catastrophic forgetting of the broad knowledge encoded in the original pretrained model. Direct use of such specialized models as RL actors can lead to suboptimal performance due to this knowledge degradation. To address this limitation, we perform a second-stage merging between the merged SFT model and the original pretrained model:

$$\mathbf{M}^{\text{BAI}} = \alpha \cdot \mathbf{M}^{\text{base}} + \beta \cdot \mathbf{M}_{\text{merge}}^{\text{sft}} \tag{2}$$

where $\alpha \in [0, 1]$ ratio and $\beta = (1 - \alpha)$ ratio represent the merging weight that controls the balance between pretrained knowledge and instruction-following capabilities, $\mathbf{M}_{\text{merge}}^{\text{sft}}$ is the merged

| Method | Benchmarks | | | | | | | | | Overall |
|---|---|---|---|---|---|---|---|---|---|---|
| | MMLU Pro | MMLU | SuperGPQA | LiveBench | MixEval-Hard | ArenaHard | AIME 2024 | MATH | MBPP+ | |
| Paradigm 1 | 67.8 | 80.8 | 38.1 | 44.4 | 48.5 | 15.4 | 18.7 | 80.9 | 67.5 | 51.3 |
| Paradigm 2 | 69.7 | 82.0 | 40.7 | 42.3 | 50.0 | 34.6 | 17.3 | 77.6 | 67.5 | 53.6 |
| Paradigm 3 | 69.2 | 80.8 | 40.5 | 43.0 | 51.5 | 16.0 | 17.7 | 77.5 | 67.2 | 51.5 |
| **BAI** | **70.2** | **82.7** | 40.6 | **44.9** | **50.8** | **35.9** | **21.3** | **81.0** | **69.3** | **55.2** |

Table 1: Performance comparison across different paradigms and the proposed BAI approach. Best results are highlighted in **blue** (BAI) and pink (other methods).

SFT model from the previous step, $\mathbf{M}^{\text{base}}$ is the original pretrained model, and $\mathbf{M}^{\text{BAI}}$ serves as our proposed RL actor initialization. The two-stage design addresses key challenges in the third paradigm: (1) Stage 1 integrates complementary capabilities from different fine-tuning approaches while maintaining parameter compatibility; (2) Stage 2 preserves the rich factual knowledge and linguistic capabilities of the pretrained model, preventing catastrophic forgetting that commonly occurs during intensive fine-tuning; (3) The parameterized control through $\alpha$ and $\beta$ provides interpretable trade-off management and control between knowledge retention and behavioral adaptation.

## 4 EXPERIMENTS AND ANALYSIS

### 4.1 IMPLEMENTATION DETAILS

In this work, we conducted RLHF experiments on MoE-2.5B/25B models. These models are scaled variants of open-source OLMoE architecture (Muennighoff et al., 2024) with augmented training parameters. For PPO-based RLHF, we used AdamW as the optimizer, setting both the actor model and critic model learning rates to $1 \times 10^{-6}$. The learning rate employed a warmup-constant scheduler. The global batch size was $4096$, with each prompt sampled once, and the mini-batch size set to $512$. The actor model was initialized using our BAI approach. The critic model was initialized using a reward model, with the GAE $\lambda$ set to $0.95$ and $\gamma$ set to $1.0$. The training utilized distributed computing across $8$ nodes with a total of $64$ GPUs. The training incorporated advanced optimization techniques including Megatron (Shoeybi et al., 2019) parallelism and Flash Attention (Dao et al., 2022), etc. Most RL experiments were trained for $1600$ steps, except for Paradigm 3 without BAI and Paradigm 3 with BAI ($\alpha = 0.6, \beta = 0.4$ merging ratio), which were trained for $3000$ steps.

### 4.2 PERFORMANCE COMPARISON ACROSS PARADIGMS

We evaluate the effectiveness of our BAI approach by comparing it against the three paradigms capabilities in language models. All RLHF experiment evaluations are conducted based on the 1600-step checkpoint to ensure fair comparison. Table 1 presents comprehensive evaluation results across diverse benchmarks spanning knowledge reasoning (MMLU Pro (Wang et al., 2024), MMLU (Hendrycks et al., 2020)), question answering (SuperGPQA (Du et al., 2025), LiveBench (White et al., 2024), MixEval-Hard (Ni et al., 2024)), conversational ability (ArenaHard-Gemini as Judge (Li et al., 2024)), mathematical reasoning (AIME 2024 (Mathematical Association of America, 2024), MATH (Hendrycks et al., 2021)), and code generation (MBPP+ (Austin et al., 2021)). BAI demonstrates superior performance compared to all three paradigms, achieving the highest overall score of $55.2$, representing a significant improvement over the best individual paradigm (Paradigm 2 at $53.6$).

The ArenaHard results reveal particularly interesting patterns when using Gemini as the judge. Paradigms 1 and 3 receive notably low scores ($15.4$ and $16.0$ respectively), while Paradigm 2 and BAI achieve substantially higher scores ($34.6$ and $35.9$). This disparity suggests that judgments tend to favor models that exhibit clear reasoning patterns and coherent response generation. Paradigm 1, lacking extensive reasoning training, produces responses that appear less structured to the judge. Paradigm 3, despite having reasoning capabilities, suffers from the sequence length collapse and training instabilities that compromise response quality and coherence. In contrast, Paradigm 2

| $\alpha \cdot \mathbf{M}^{\text{base}} + \beta \cdot \mathbf{M}^{\text{sft}}_{\text{merge}}$ | $\alpha = 0.1$ $\beta = 0.9$ | $\alpha = 0.2$ $\beta = 0.8$ | $\alpha = 0.3$ $\beta = 0.7$ | $\alpha = 0.4$ $\beta = 0.6$ | $\alpha = 0.5$ $\beta = 0.5$ | $\alpha = 0.6$ $\beta = 0.4$ | $\alpha = 0.7$ $\beta = 0.3$ | $\alpha = 0.8$ $\beta = 0.2$ | $\alpha = 0.9$ $\beta = 0.1$ |
|---|---|---|---|---|---|---|---|---|---|
| MMLU Pro | 70.2 | 69.7 | 69.8 | 69.4 | **70.7** | 68.3 | 68.6 | 68.2 | 68.1 |
| MMLU | **82.7** | 81.3 | 81.1 | 82.1 | 80.9 | 81.7 | 81.8 | 80.2 | 81.3 |
| SuperGPQA | 40.6 | **40.8** | 39.0 | 40.2 | 39.6 | 39.4 | 37.6 | 38.7 | 38.9 |
| LiveBench | 44.9 | 44.6 | 44.0 | 44.3 | 45.0 | 44.6 | **45.5** | 44.8 | 44.3 |
| MixEval-Hard | 50.8 | 50.6 | 50.8 | 49.9 | 48.7 | 51.0 | **51.2** | 50.0 | 47.8 |
| ArenaHard | **35.9** | 33.3 | 29.9 | 27.0 | 25.2 | 20.7 | 16.0 | 20.3 | 11.5 |

Table 2: Performance evaluation across different merging ratios. Best results are highlighted in **blue**.

maintains stable reasoning patterns, while BAI not only preserves these patterns but enhances them through balanced initialization, resulting in the highest ArenaHard score.

BAI shows notable improvements in mathematical reasoning tasks, achieving the highest scores on both AIME 2024 and MATH benchmarks compared to all other paradigms. These improvements demonstrate that BAI successfully preserves and enhances reasoning capabilities while avoiding the degradation typically observed in Paradigm 3. Across most benchmarks, BAI either matches or exceeds the performance of individual paradigms, achieving the highest scores in these benchmarks, indicating the robustness and generalizability of our approach across different task domains.

### 4.3 Performance Evaluation Across BAI Configurations

Table 2 presents the evaluation of performance across different BAI merging configurations, revealing key insights into the optimal balance between pretrained knowledge and specialized reasoning capabilities. The analysis demonstrates a clear trend: configurations with higher SFT weights (lower $\alpha$ values) consistently achieve superior performance across most benchmarks. The superior performance of SFT-heavy configurations stems from their preservation of reasoning capabilities and instruction-following behaviors acquired during distillation-based reasoning fine-tuning. This advantage is particularly pronounced in benchmarks such as MMLU Pro (70.2%), MMLU (82.7%), and ArenaHard (35.9%), which demand structured problem-solving approaches that align closely with chain-of-thought methodologies. The consistent top-tier performance of the $(\alpha = 0.1, \beta = 0.9)$ configuration across diverse evaluation metrics underscores the effectiveness of prioritizing specialized reasoning patterns over raw pretrained knowledge.

Moreover, configurations with higher pretrain weights ($\alpha \geq 0.6$) also demonstrate competitive performance on specific tasks. For instance, certain benchmarks like MixEval-Hard achieve peak performance at $(\alpha = 0.7, \beta = 0.3)$ (51.2%), suggesting that the broad knowledge base from pretraining remains valuable for tasks requiring extensive factual recall and general linguistic competence. This task-dependent behavior indicates that the optimal merging ratio may vary based on the specific cognitive demands of different evaluation scenarios.

The ArenaHard benchmark reveals the most dramatic sensitivity to merging ratios, with performance declining precipitously from 35.9% to 11.5%. This steep degradation highlights the fundamental importance of instruction-following and conversational reasoning capabilities that are primarily encoded in the SFT component. The results suggest that while pretrained knowledge provides a foundation, the specialized behavioral patterns learned during distillation-based reasoning fine-tuning are indispensable for complex interactive reasoning tasks in the third paradigm.

### 4.4 Reward Hockey Stick Curve Phenomenon in Paradigm 3

Paradigm 3 faces a critical challenge in the form of reward model instability during RLHF training. When models undergo extensive distillation-based reasoning fine-tuning and are subsequently used as both the reward model and RL actor, we observe the Reward Hockey Stick Curve phenomenon characterized by a "Hockey Stick"-shaped trajectory in reward scores.

Figure 2 and Figure 6 illustrates this phenomenon across different sequence length ranges during RLHF training. As shown by the pink curves, the Reward Hockey Stick Curve exhibits three distinct phases: an initial decline where RM scores decrease from their starting values after distillation-based reasoning fine-tuning, a trough phase where performance plateaus at the lowest point, and a recovery phase featuring gradual improvement that often surpasses the original performance levels.

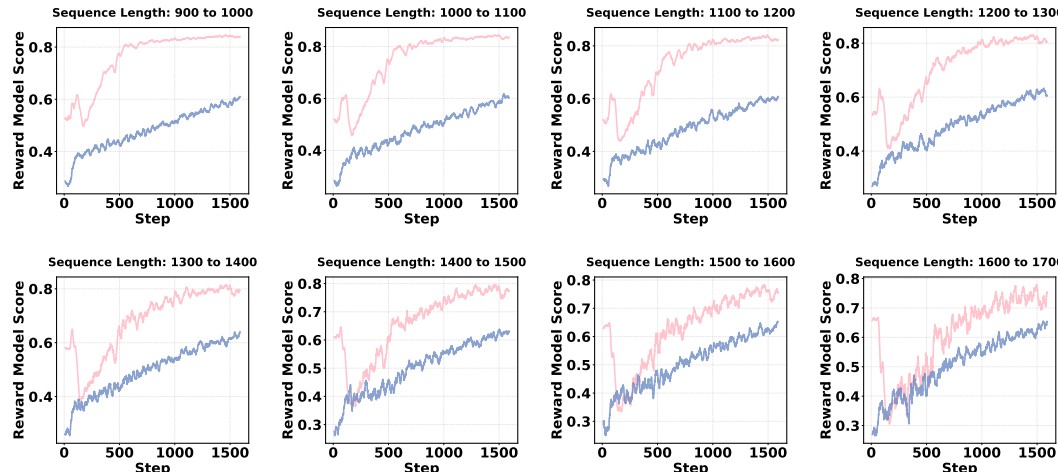

Figure 2: Reward Hockey Stick Curve phenomenon for different generated sequence length. The pink curve represents Paradigm 3 without BAI, while the blue curve represents Paradigm 3 with BAI. Each figure represents the reward scores of samples with generated lengths in the interval from $i100$ to $(i+1)100$, where i represents integers ranging from 9 to 16.

This phenomenon stems from the fundamental mismatch between the specialized reasoning patterns learned during distillation-based fine-tuning and the requirements of RL optimization. The intensive parameter updates during reasoning fine-tuning create optimization landscapes that are highly sensitive to distribution shifts, leading to reward signal instability when transitioning from supervised to RL-generated samples. This mismatch often triggers reward hacking behaviors, where the model learns to exploit the reward signal in unintended ways, initially achieving higher scores through shortcuts rather than genuine reasoning improvement. This instability undermines training consistency and represents a core barrier to successful implementation of the third paradigm. The Hockey Stick Curve motivates our BAI approach. BAI addresses the underlying causes of reward instability and reward hacking tendencies, enabling stable training dynamics from the onset of RL training.

### 4.4.1 MITIGATING THE REWARD HOCKEY STICK CURVE THROUGH BAI

To address the Reward Hockey Stick Curve phenomenon in the third paradigm, we demonstrate how our BAI approach effectively mitigates this critical training instability. To validate our method's effectiveness, we employ a balanced ($\alpha = 0.5, \beta = 0.5$) merging ratio between the pretrained model and the distillation-based reasoning SFT model, deliberately chosen to demonstrate robustness without extensive hyperparameter optimization.

As shown by the blue curves in Figure 2 and Figure 6, BAI approach substantially mitigates the Reward Hockey Stick Curve phenomenon, maintaining stable training trajectories. The effectiveness of our approach stems from several key factors: BAI creates a more balanced parameter distribution that reduces extreme specialization, preserves broad knowledge from pretraining that enhances stability, exhibits more stable gradient flows during early RL training, and reduces overfitting to specific reasoning patterns. Importantly, the balanced initialization effectively mitigates reward hacking behaviors by providing more robust starting points that are less susceptible to exploiting reward signal shortcuts. The consistent performance across different sequence length ranges indicates that BAI addresses fundamental training dynamics rather than superficial symptoms, while the simple ratio demonstrates the method's practical viability.

To further investigate the influence of different configurations for our BAI approach, we conduct a comprehensive comparison across different merging ratios. Figure 7 presents the reward score trajectories for three BAI configurations: ($\alpha = 0.1, \beta = 0.9$) (red curves), ($\alpha = 0.5, \beta = 0.5$) (blue curves), and ($\alpha = 0.9, \beta = 0.1$) (grey curves), analyzed across different sequence length ranges. The results reveal distinct performance patterns across different sequence length ranges. For shorter sequences, all three configurations demonstrate relatively stable performance, with the balanced ($\alpha = 0.5, \beta = 0.5$) configuration showing slight advantages in convergence speed and final

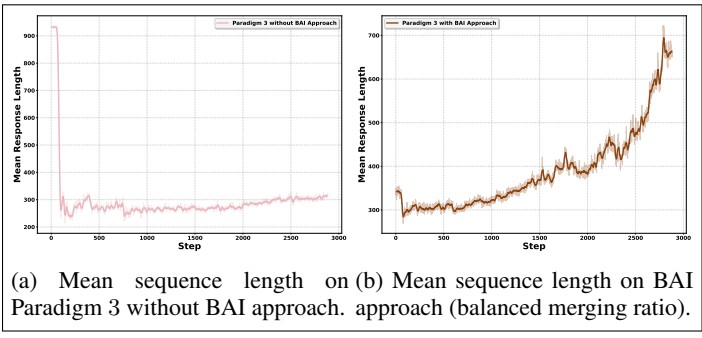 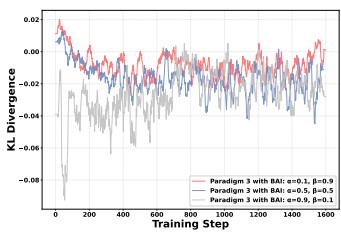

(a) Mean sequence length on Paradigm 3 without BAI approach.  (b) Mean sequence length on BAI approach (balanced merging ratio).  (c) KL divergence stability analysis across BAI configurations.

Figure 3: Left figures represent sequence length collapse mitigation comparing Paradigm 3 without BAI vs. with BAI. Right figure represents KL divergence stability analysis with BAI approach.

reward scores. However, as sequence length increases, the differences become more pronounced. The SFT-heavy configuration ($\alpha = 0.1, \beta = 0.9$) consistently achieves the highest reward scores across longer sequence ranges, demonstrating superior performance for complex reasoning tasks that require extended chain-of-thought generation. This advantage becomes particularly evident in the long sequence ranges, where the red curves consistently outperform other configurations by substantial margins. In contrast, the pretrain-heavy configuration ($\alpha = 0.9, \beta = 0.1$) shows more conservative performance gains, with grey curves typically plateauing at lower reward levels. While this configuration provides stability, it appears to sacrifice some reasoning capability for robustness, particularly in longer sequence generation tasks. The balanced ($\alpha = 0.5, \beta = 0.5$) configuration achieves a middle ground, demonstrating competitive performance across all sequence lengths while maintaining training stability. This configuration represents an optimal choice for balancing reasoning capability with training robustness, confirming the effectiveness of balanced merging ratios.

### 4.5 ADDRESSING SEQUENCE LENGTH COLLAPSE THROUGH BAI

The paradigm 3 faces another critical challenge in the form of Sequence Length Collapse, where models experience dramatic reduction in sample generation during early RL training phases. As illustrated in Figure 3a, models initialized from pure distillation-based reasoning fine-tuned models exhibit catastrophic drops in generated sequence length within the first few training steps, with no recovery throughout the entire training process. This phenomenon fundamentally undermines the model's ability to generate comprehensive reasoning chains and detailed responses. The sequence length collapse occurs due to the distributional mismatch between specialized reasoning patterns learned during distillation-based fine-tuning and RL optimization requirements. When transitioning to RL training, the reward model optimization creates tension between the specialized SFT patterns and reward signal expectations. This mismatch triggers an over-correction mechanism where the model rapidly shortens responses to achieve higher immediate rewards, effectively engaging in reward hacking by producing concise responses that score well but lack reasoning depth. The concentrated parameter updates during distillation-based reasoning fine-tuning create brittle optimization landscapes that are susceptible to rapid degradation under RL gradient updates.

To address this challenge, we evaluate our BAI approach across different merging ratios, examining how the balance between pretrained and SFT parameters affects sequence length stability. Figure 4 presents the mean sequence lengths for different merging ratios during RL training. The results reveal consistent behavior: every merging ratio effectively reduces or eliminates the initial sequence length collapse compared to the pure SFT baseline. More importantly, ratios closer to the balanced ($\alpha = 0.5, \beta = 0.5$) configuration exhibit the most desirable behavior—not only do they prevent the initial collapse, but they also demonstrate progressive sequence length growth throughout training.

This progressive improvement in balanced merging ratios can be attributed to the optimal equilibrium between knowledge preservation and reasoning capability. Ratios heavily weighted toward the pretrained model (e.g., ($\alpha = 0.9, \beta = 0.1$)) lack sufficient reasoning initialization, requiring longer training to develop CoT capabilities. Conversely, ratios favoring the SFT model (e.g., ($\alpha = 0.1, \beta = 0.9$)) retain more reasoning patterns but inherit instability from the specialized fine-tuning. The balanced ($\alpha = 0.5, \beta = 0.5$) and ($\alpha = 0.6, \beta = 0.4$) ratios achieve an optimal

compromise, preserving enough pretrained stability to prevent collapse while maintaining sufficient reasoning capability to enable progressive development.

Importantly, the sequence length growth observed in balanced ratios highlights that optimal model development requires consideration of both performance metrics and training dynamics. While SFT-heavy configurations may achieve higher immediate benchmark scores, the progressive sequence length improvement in balanced ratios demonstrates healthier learning patterns that are more likely to sustain long-term capability development. This finding underscores that effective reasoning model training should not solely pursue metric optimization but must also ensure stable and progressive training states. To further validate the long-term sequence length growth of our approach, we extended training for the ($\alpha = 0.6, \beta = 0.4$) configuration by an additional $1,400$ steps. As illustrated in Figure 3b, sequence length continues to grow progressively with training steps. This demonstrates that our BAI approach not only prevents initial collapse but also establishes a foundation for sustained capability development, suggesting that longer training could yield substantial improvements in reasoning depth and comprehensiveness. This balance creates a more robust optimization landscape that supports both immediate stability and long-term capability growth, establishing the foundation for effective reasoning model development in the third paradigm.

### 4.6    KL Divergence Analysis of BAI Configurations

To better understand the training dynamics enabled by our BAI approach, we analyze the KL divergence between the training policy and the sampling policy throughout the RL training process. The KL divergence serves as a critical indicator of policy stability during optimization, with effective training characterized by controlled divergence patterns that balance adaptation with stability.

Figure 3c presents the KL divergence trajectories for three representative BAI configurations, revealing distinct behavioral patterns: SFT-heavy ($\alpha = 0.1, \beta = 0.9$) demonstrates the most stable patterns with minimal fluctuations, reflecting consistent policy behavior from specialized reasoning patterns. Pretrain-heavy ($\alpha = 0.9, \beta = 0.1$) configuration exhibits the most volatile pattern with frequent spikes, indicating challenges in adapting broad pretrained parameters to specific RL objectives. The balanced ($\alpha = 0.5, \beta = 0.5$) configuration achieves a middle ground with moderate fluctuations that indicate healthy optimization dynamics while maintaining better stability than the pretrain-heavy setup. These KL divergence patterns provide additional evidence supporting our BAI approach, demonstrating that balanced merging ratios enable effective adaptation to reward signals while maintaining training stability. More detailed analysis can be found in Section A.3.

## 5    Conclusion

In this paper, we investigate the development of reasoning capabilities in large language models through three established paradigms: the instruction tuning and alignment paradigm (Paradigm 1), distillation-based reasoning fine-tuning (Paradigm 2), and their combination through applying RLHF to distillation-trained models (Paradigm 3). Although Paradigms 1 and 2 have proven effective independently, Paradigm 3 faces critical training instabilities. Our analysis identified two fundamental challenges in Paradigm 3: Sequence Length Collapse, where models experience dramatic reduction in language generation during early RL training, and the Reward Hockey Stick Curve, featuring initial reward score degradation followed by gradual recovery. These phenomena fundamentally compromise the model's ability to maintain detailed reasoning chains and stable training dynamics. To address these challenges, we proposed Balanced Actor Initialization (BAI), a two-stage weighted model merging approach. BAI creates robust actor initializations that prevent training instabilities while maintaining specialized reasoning capabilities. Our experimental evaluation across diverse benchmarks demonstrates BAI's effectiveness. BAI outperforms all three individual paradigms while maintaining stable training throughout the RL process. The approach consistently eliminates sequence length collapse, mitigates reward curve instabilities, and enables continuous sequence length improvement during training. These results confirm that BAI successfully enables stable training in Paradigm 3, allowing practitioners to leverage both the efficiency of distillation and the alignment benefits of reinforcement learning from human feedback optimization.

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

# A APPENDIX

## A.1 USAGE OF LLM.

We utilized ChatGPT to improve the manuscript's readability.

## A.2 MEAN RESPONSE LENGTH ACROSS BAI MERGING RATIOS

This section provides comprehensive empirical evidence for the sequence length behaviors observed across different BAI merging ratios during RL training. Figure 4 presents the complete experimental results for nine different merging configurations, ranging from heavily pretrained-weighted ($\alpha = 0.1, \beta = 0.9$) to heavily SFT-weighted ($\alpha = 0.9, \beta = 0.1$) ratios.

**Pretrained-Heavy BAI Configurations:** Models with ratios ($\alpha = 0.1, \beta = 0.9$), ($\alpha = 0.2, \beta = 0.8$), and ($\alpha = 0.3, \beta = 0.7$) exhibit the most dramatic sequence length growth patterns. These configurations begin training with relatively shorter initial sequences but demonstrate steep, sustained growth throughout the training process. The ($\alpha = 0.1, \beta = 0.9$) configuration shows the most aggressive growth. This behavior reflects the model's gradual acquisition of reasoning capabilities from a more general pretrained foundation.

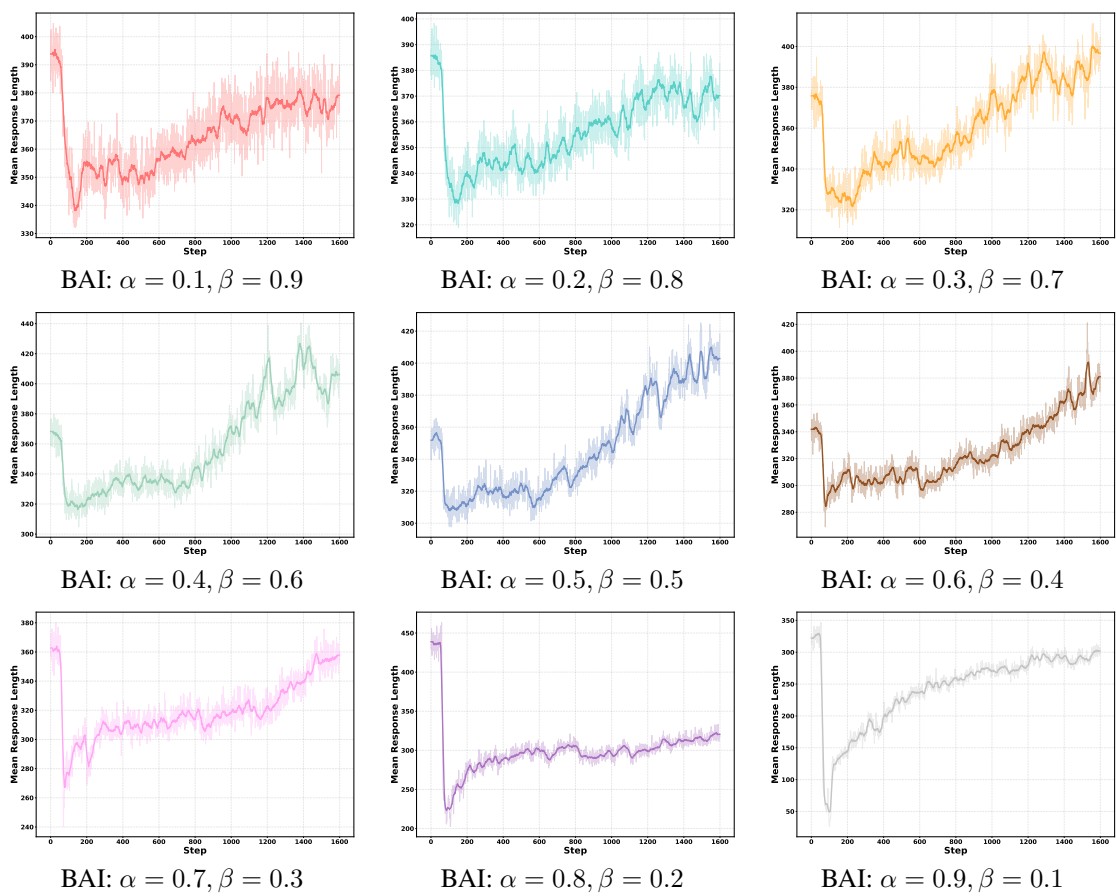

Figure 4: Mean sequence length on different BAI merging ratio.

**Balanced BAI Configurations:** The balanced ratios ($\alpha = 0.4, \beta = 0.6$), ($\alpha = 0.5, \beta = 0.5$), and ($\alpha = 0.6, \beta = 0.4$) demonstrate the most stable and progressive sequence length development. These configurations successfully avoid the catastrophic collapse observed in pure SFT baselines while maintaining steady upward trajectories. Notably, the ($\alpha = 0.6, \beta = 0.4$) configuration ex-

hibits particularly smooth growth with minimal variance, suggesting optimal stability in the optimization landscape.

**SFT-Heavy BAI Configurations:** Configurations with higher SFT weighting, including ($\alpha = 0.7, \beta = 0.3$), ($\alpha = 0.8, \beta = 0.2$), and ($\alpha = 0.9, \beta = 0.1$), show progressively diminished sequence length growth. The ($\alpha = 0.9, \beta = 0.1$) configuration demonstrates the most concerning behavior, with sequence lengths remaining relatively flat throughout training. This pattern indicates that heavy reliance on SFT parameters introduces instabilities that inhibit natural sequence length development during RL optimization.

The variance patterns across different configurations provide additional insights into training stability. Pretrained-heavy configurations exhibit higher variance during early training phases, which gradually stabilizes as the model develops reasoning capabilities. In contrast, balanced configurations maintain consistent variance throughout training, indicating more stable optimization dynamics. SFT-heavy configurations show increasing variance in later training phases, suggesting potential optimization instabilities that could lead to training degradation.

The progressive sequence length growth observed in pretrained-heavy and balanced configurations suggests that these merging strategies create optimization landscapes conducive to sustained capability development. The steep growth curves in configurations like ($\alpha = 0.1, \beta = 0.9$) and ($\alpha = 0.2, \beta = 0.8$) indicate that extended training could yield substantial improvements in reasoning depth and comprehensiveness. Conversely, the behavior in SFT-heavy configurations demonstrates inherent limitations in their capacity for continued development. This finding has important implications for computational resource allocation, as extending training for these configurations may yield diminishing returns compared to more balanced approaches.

### A.3 ADDITIONAL ANALYSIS OF KL DIVERGENCE IN BAI TRAINING

To better understand the training dynamics enabled by our BAI approach, we analyze the KL divergence between the training policy and the sampling policy throughout the RL training process. The KL divergence serves as a critical indicator of policy stability during optimization, with effective training typically characterized by controlled divergence patterns that avoid both excessive drift and stagnation.

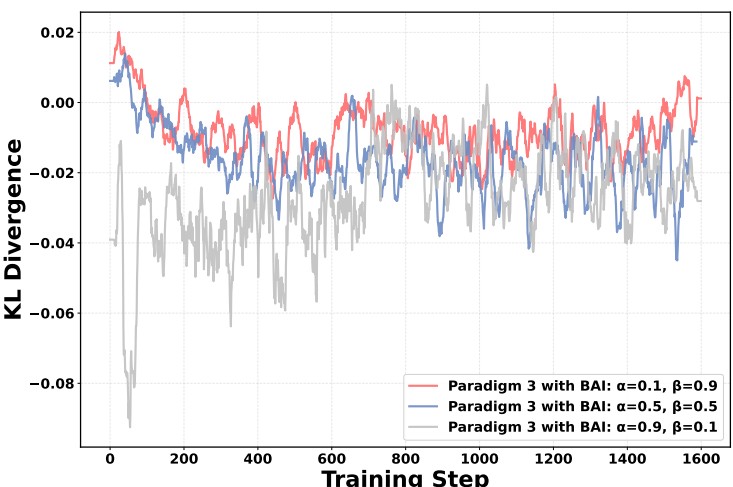

Figure 5: KL divergence analysis across BAI configurations.

Figure 5 presents the KL divergence trajectories for three representative BAI configurations: ($\alpha = 0.1, \beta = 0.9$) (SFT-heavy), ($\alpha = 0.5, \beta = 0.5$) (balanced), and ($\alpha = 0.9, \beta = 0.1$) (pre-train heavy). The results reveal distinct behavioral patterns that correlate with the merging characteristics and provide insight into the underlying optimization dynamics. The BAI ($\alpha = 0.1, \beta = 0.9$) configuration demonstrates the most stable KL divergence patterns with minimal fluctuations throughout

training. This stability stems from the specialized reasoning patterns acquired during fine-tuning, which provide consistent policy behavior under RL optimization.

In contrast, the $(\alpha = 0.9, \beta = 0.1)$ configuration exhibits the most volatile KL divergence pattern with frequent spikes and irregular fluctuations. This instability reflects the challenge of adapting the heavy pretrained parameters to specific RL objectives, where the broad parameter configuration struggles to maintain consistent policy behavior.

The balanced BAI $(\alpha = 0.5, \beta = 0.5)$ configuration achieves a great ground, exhibiting moderate KL divergence fluctuations that indicate healthy optimization dynamics. While showing more variation than the SFT-heavy configuration, it maintains better stability than the pretrain-heavy setup, demonstrating effective adaptation while preserving training consistency.

These KL divergence patterns provide additional evidence supporting our BAI approach and help explain the superior performance of balanced configurations. The controlled divergence in balanced merging ratios indicates that BAI enables effective adaptation to reward signals while maintaining training stability, establishing favorable conditions for sustained learning and capability growth.

## A.4 FUTURE DIRECTIONS

Although our BAI approach demonstrates significant effectiveness in addressing training instabilities in the third paradigm, several directions warrant further investigation. Future work could explore adaptive merging strategies that dynamically adjust weights during training, investigate the application of similar approaches to other model architectures and training objectives to better understand the optimization dynamics underlying these phenomena. Additionally, extending our analysis to other specialized fine-tuning domains beyond reasoning, such as agent training and multi-modalities, could validate the broader applicability of weighted model merging strategies.

The success of BAI establishes that proper actor initialization is fundamental to stable training in the third paradigm. By demonstrating that strategic model weight interpolation can create robust initializations that prevent training instabilities, this work highlights the critical importance of initialization strategies in modern language model development. Our findings open new avenues for developing initialization methodologies that enable reasoning models to successfully leverage both distillation efficiency and human feedback optimization while maintaining training stability throughout the process.

## A.5 RELATED WORK ON MODEL MERGING

Model merging has emerged as a powerful technique for combining the capabilities of multiple trained models without requiring additional training data or computational resources. Solar (Kim et al., 2023) demonstrated that simple weight averaging could improve model performance across different domains. This foundational approach has since been extended to more sophisticated merging strategies.

Recent advances in model merging include task-specific weight interpolation (Matena & Raffel, 2022; Kim et al., 2023), where models trained on different tasks are combined to create multi-capable systems. The TIES-Merging approach (Yadav et al., 2024) addresses sign conflicts and magnitude differences when merging models with overlapping capabilities. Fisher-weighted averaging (Matsuoka, 2022) leverages Fisher information to determine optimal combining weights, while SLERP-based approaches (Goddard et al., 2024) use spherical linear interpolation for smoother parameter transitions.

More recently, evolutionary and optimization-based merging methods have gained attention. Recent works (Akiba et al., 2024; Li et al., 2025b) introduced algorithms for discovering optimal merging configurations, while DARE (Tang et al., 2024) addresses redundant parameters during merging. These approaches demonstrate that model merging can achieve performance comparable to or exceeding individual specialized models across diverse tasks.

Different from all these works, which primarily focus on merging models to create multi-capable systems for inference, our work addresses this gap by specifically examining model merging as an effective approach for creating robust initializations for reinforcement learning training, particularly in the context of reasoning model development.

## B    Reward Hockey Stick Curve Phenomenon

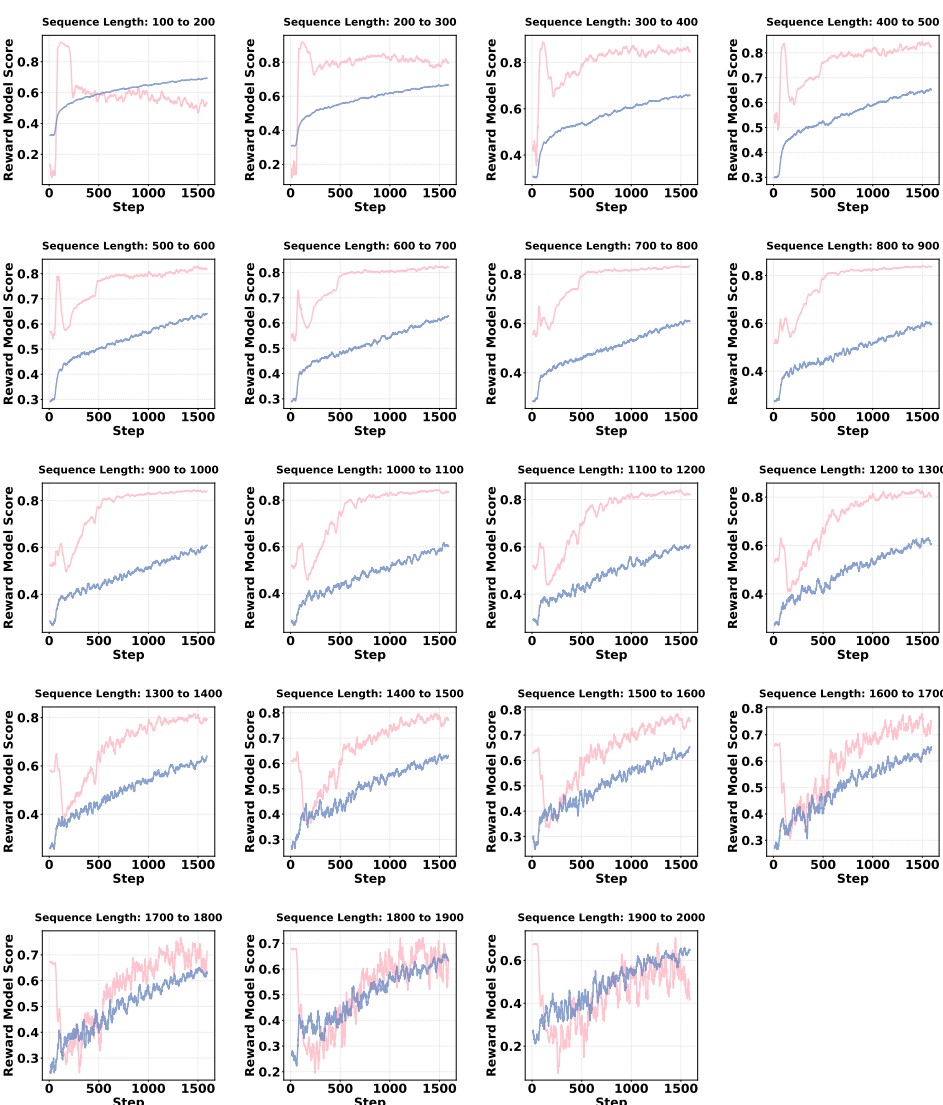

Figure 6: Reward Hockey Stick Curve phenomenon for different generated sequence length. The pink curve represents Paradigm 3 without BAI, while the blue curve represents Paradigm 3 with BAI. Each figure represents the reward scores of samples with generated lengths in the interval from i×100 to (i+1)×100, where i represents integers ranging from 0 to 19.

### B.1    More Details about Reward Hockey Stick Curve Phenomenon

Figure 6 provides more observations of the reward Hockey Stick Curve phenomenon across three BAI configurations and all sequence length ranges. The red curves ($\alpha = 0.1, \beta = 0.9$) consistently demonstrate the most stable reward trajectories, avoiding the characteristic initial decline and maintaining steady improvement throughout training. This stability is particularly evident in longer sequences where other configurations show significant volatility.

The blue curves ($\alpha = 0.5, \beta = 0.5$) exhibit moderate Hockey Stick effects in shorter sequences but achieve increasingly stable performance as sequence length increases. The grey curves ($\alpha = 0.9, \beta = 0.1$) display the most pronounced Hockey Stick patterns, with severe initial declines and prolonged recovery phases across most sequence ranges.

The systematic analysis across all sequence length intervals reveals important insights into the relationship between merging ratios and training stability. In shorter sequences (100-800 tokens), all configurations eventually converge to similar final performance levels, but the training trajectories differ significantly in stability and convergence speed. For longer sequences (900+ tokens), the performance gaps become more pronounced, with SFT-heavy configurations maintaining substantial advantages throughout training. This pattern suggests that the benefits of preserving reasoning patterns through higher SFT weighting become increasingly critical as task complexity grows.

These detailed results confirm that SFT-heavy BAI configurations effectively mitigate the Hockey Stick phenomenon while maintaining superior final performance, particularly for complex reasoning tasks requiring longer sequences. The systematic variation across merging ratios validates our approach and provides clear guidance for practical implementation based on specific sequence length requirements.

## B.2    REWARD SCORE CURVES ACROSS BAI RATIOS

This section presents a comprehensive analysis of reward score trajectories across different BAI merging ratios and sequence length ranges. Figure 7 compares three representative BAI configurations: $(\alpha = 0.1, \beta = 0.9)$ (red curves), $(\alpha = 0.5, \beta = 0.5)$ (blue curves), and $(\alpha = 0.9, \beta = 0.1)$ (grey curves) across 20 different sequence length intervals ranging from 100 to 2000 tokens.

**SFT-Heavy Configuration** ($\alpha = 0.1, \beta = 0.9$)**:** The red curves consistently demonstrate the highest final reward scores across all sequence ranges, with this advantage becoming more pronounced as sequence length increases. In short sequences, this configuration shows marginal performance gains in the 400-500 token range. For medium sequences, the red curves demonstrate superior performance across all intervals. In long sequences, the SFT-heavy configuration achieves substantial advantages, consistently scoring 0.1-0.15 points higher than other configurations, with particularly steep upward trajectories in the 1300-2000 token ranges. This configuration exhibits robust performance scaling with sequence length, indicating that preserved reasoning patterns from distillation-based fine-tuning provide significant advantages for complex reasoning tasks requiring extended chain-of-thought generation.

**Balanced Configuration** ($\alpha = 0.5, \beta = 0.5$)**:** The blue curves represent an optimal balance between performance and stability across all sequence ranges. In short sequences, this configuration shows slight advantages in convergence stability with rapid initial convergence within the first 500 training steps. For medium sequences, the balanced configuration maintains competitive performance while showing more stable training dynamics with smoother curve trajectories. In long sequences, while not achieving the highest scores, this configuration demonstrates the most consistent training dynamics with smooth convergence patterns and reduced variance, making it suitable for applications prioritizing training reliability.

**Pretrain-Heavy Configuration** ($\alpha = 0.9, \beta = 0.1$)**:** The grey curves show the most conservative performance profile across all sequence ranges. This configuration maintains similar performance to other configurations in short sequences, but shows consistent but limited improvements in medium sequences. In long sequences, this configuration exhibits signs of plateauing and sacrifices reasoning capability, particularly evident in longer sequence generation tasks. However, it provides excellent training stability throughout all ranges, making it suitable for applications prioritizing training stability over peak performance.

The analysis reveals important trade-offs for practical applications. For tasks requiring maximum reasoning performance on complex, long-form problems, the SFT-heavy configuration provides optimal results, particularly for sequences exceeding 1000 tokens. For general-purpose applications requiring balanced performance and training stability, the balanced configuration offers the best compromise. For applications prioritizing training stability over peak performance, the pretrain-heavy configuration provides the most conservative approach. The progressive divergence in performance as sequence length increases demonstrates that merging ratio selection becomes increasingly critical for applications requiring long-form reasoning, while the choice matters less for shorter sequence tasks.

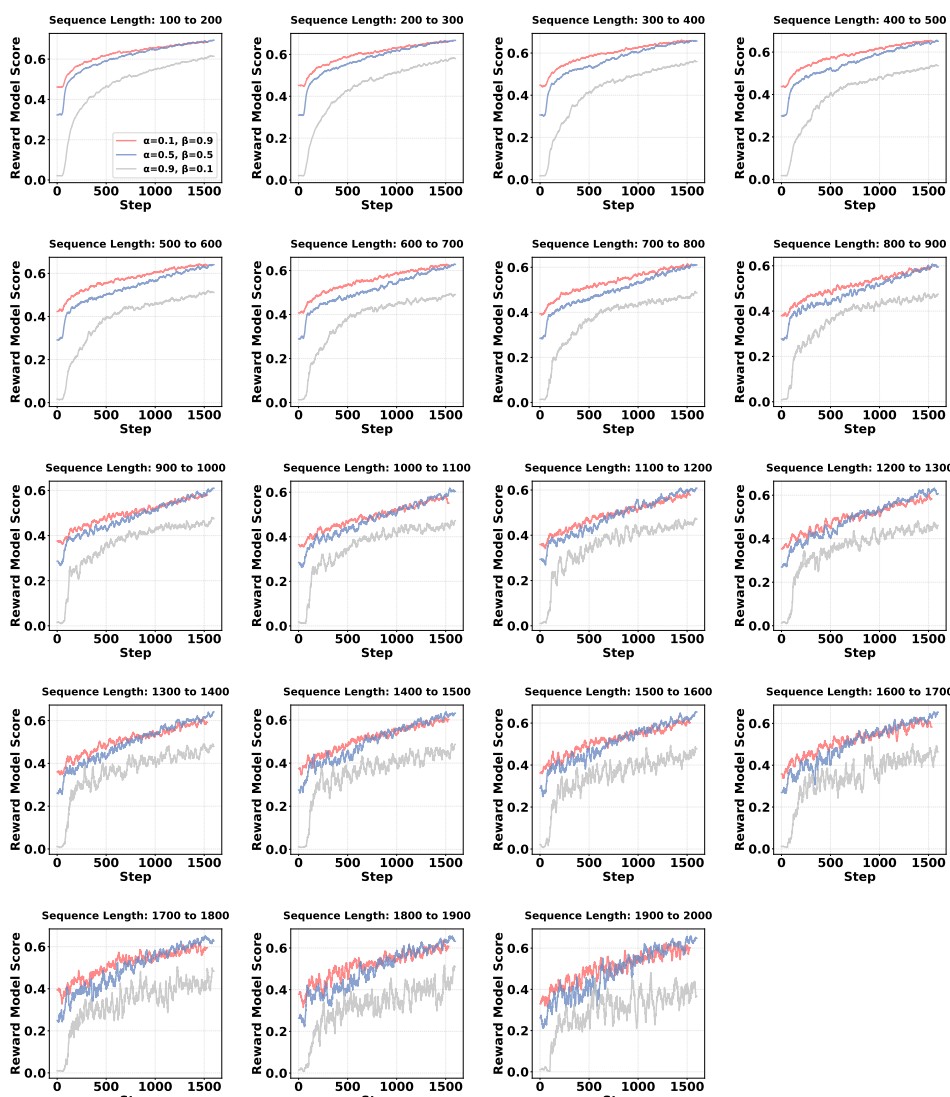

Figure 7: Reward score curve across different BAI ratio. The red curve represents Paradigm 3 BAI ($\alpha = 0.1, \beta = 0.9$), the blue curve represents Paradigm 3 with BAI ($\alpha = 0.5, \beta = 0.5$) , while the grey curve represents Paradigm 3 with BAI ($\alpha = 0.9, \beta = 0.1$). Each figure represents the reward scores of samples with generated lengths in the interval from i×100 to (i+1)×100, where i represents integers ranging from 0 to 19.

## C  CASE STUDY

To illustrate the qualitative differences underlying our quantitative evaluation results, we present representative examples comparing model responses across paradigms. The cases demonstrate distinct reasoning characteristics: Paradigm 1 shows basic analytical capability but lacks precision, often containing conceptual errors while arriving at correct answers through mechanical steps. Paradigm 2 exhibits structured reasoning with clear organization and verification steps, but can be overly rigid and miss nuanced distinctions. Paradigm 3 with BAI produces the most sophisticated responses, demonstrating deep theoretical understanding, comprehensive analysis, and practical applicability. These concrete examples validate our performance findings, showing how BAI-enhanced Paradigm 3 achieves superior reasoning quality across diverse domains - from philosophical theory comprehension to mathematical problem-solving - explaining its highest benchmark performance and robust generalization across task types.

| Prompt Label: A | According to Stevenson, moral disagreements involve:
A) disagreement in interest.
B) disagreement in belief.
C) disagreement in theory.
D) disagreement in practice. |
|---|---|
| Paradigm 1 | 1. Analyze each option:
Option A: Disagreement in interest typically refers to differences in what people want or desire. In the context of moral disagreements, moral issues are not primarily about different interests. For example, if two people are arguing about whether it is wrong to lie to save a life, it is not mainly a matter of different interests they have. So, option A is incorrect.
Option B: Stevenson argues that moral disagreements involve disagreement in belief. When people have moral disagreements, they often have different beliefs about what is right and wrong. For instance, one person may believe that euthanasia is always wrong based on religious beliefs, while another person may believe that under certain circumstances, euthanasia is morally acceptable. This difference is a disagreement in belief about the moral status of euthanasia. So, option B is correct.
Option C: Disagreement in theory is too broad and not the core aspect of what Stevenson emphasizes in moral disagreements. The focus is on the underlying beliefs about moral values rather than differences in theoretical frameworks in general. So, option C is incorrect.
Option D: Disagreement in practice may seem related, but the key aspect of moral disagreements according to Stevenson is the underlying disagreement in beliefs about what is morally right or wrong, not just differences in how people act in practice. So, option D is incorrect.
Answer: B |
| Paradigm 2 | To determine which option Stevenson identifies as involved in moral disagreements, let's recall Stevenson's views on ethical disagreement. Stevenson argued that moral disagreements primarily stem from differences in attitudes rather than just differences in beliefs.
He distinguished between two types of disagreement: disagreement in belief (where people disagree about the facts of a situation) and disagreement in attitude (which involves differing emotional or evaluative stances). In moral disputes, Stevenson contended that the key source of conflict is not merely about factual beliefs but about conflicting desires, preferences, or values.
Option A, "disagreement in interest," might seem related, but Stevenson's focus was more on the clash of attitudes or interests rather than a specific type of interest. Option C, "disagreement in theory," is too broad and not specific to moral disagreements. Option D, "disagreement in practice," refers to actions but does not capture the underlying disagreement in attitudes that Stevenson emphasized.
Option B, "disagreement in belief," aligns with Stevenson's view that moral disagreements often involve differing factual beliefs about the consequences or nature of an action. However, he also stressed that the deeper disagreement is in attitudes. But given the options provided, the most accurate according to Stevenson's analysis would be that moral disagreements involve disagreement in belief, even though there's a broader element of disagreement in attitude.
Answer: B |
| Paradigm 3 | Robert L. Stevenson was a philosopher known for his work on emotivism, a meta-ethical theory that posits moral statements primarily express emotions or attitudes rather than objective facts. In his view, moral disagreements do not stem from differences in beliefs about factual matters. Emotivists like Stevenson argue that when people have moral disagreements, the core issue isn't that one person has a belief that the other lacks. Instead, the disagreement arises from differing attitudes or emotional responses towards a particular action or situation. For instance, if one person thinks an action is right and another thinks it's wrong, this isn't because they have different beliefs about the action's consequences or properties; rather, it's because they have contrasting emotional reactions towards it.
Thus, moral disagreements, according to Stevenson, involve disagreement in interest—different emotional or attitudinal stances towards the same object.
Options B, C, and D don't capture the essence of Stevenson's position. Disagreement in belief (B) implies a difference in factual understanding, which emotivism rejects as the primary cause of moral disputes. Disagreement in theory (C) refers to overarching philosophical frameworks, which isn't the main focus here. Disagreement in practice (D) would suggest differing application of principles, but Stevenson's view is that the disagreement is at the level of emotional response, not practical application.
Answer: A |

Table 3: Case Study: Stevenson's Theory of Moral Disagreements

| | |
|---|---|
| Prompt | I would like to build a magic the gathering deck. this deck contains 99 cards. How many lands should i play to draw on average 3 lands in my starting 7 cards hand. explain the calculus. |
| Paradigm 1 | 1. Use the hypergeometric distribution formula: In a Magic: The Gathering deck, assume the total number of cards in the deck is $N = 99$ (the size of the deck). Let the number of lands in the deck be $K$. The number of cards drawn from the deck is $n = 7$ (the starting hand size). We want the expected number of lands in the hand to be $E(X) = 3$.

The probability mass function of the hypergeometric distribution for the number of successes (lands in this case) in a sample of size $n$ drawn from a population of size $N$ containing $K$ successes is given by $P(X = k) = \frac{\binom{K}{k}\binom{N-K}{n-k}}{\binom{N}{n}}$, where $\binom{a}{b} = \frac{a!}{b!(a-b)!}$. The expected value $E(X)$ of a hypergeometric random variable $X$ is given by the formula $E(X) = n \times \frac{K}{N}$.
2. Solve for the number of lands $K$: We know that $n = 7$, $N = 99$, and $E(X) = 3$. Substituting these values into the formula $E(X) = n \times \frac{K}{N}$, we get: $3 = 7 \times \frac{K}{99}$. First, we can rewrite the equation as $3 = \frac{7K}{99}$. To solve for $K$, we multiply both sides of the equation by 99: $3 \times 99 = 7K$. So, $297 = 7K$. Then, we divide both sides by 7 to find $K = \frac{297}{7} \approx 42.43$.
Since we can't have a fraction of a land in the deck, we would typically play 42 lands in the 99-card deck to get, on average, 3 lands in the starting 7-card hand. |
| Paradigm 2 | The Gathering deck to have an average of 3 lands in your starting 7-card hand, we need to use probability and combinatorics.
**Key Concepts:** 1. Hypergeometric Distribution: This models the probability of drawing a certain number of "successes" (lands in this case) from a finite population without replacement. 2. Average (Expected Value): The expected number of lands in a 7-card hand can be calculated using the hypergeometric distribution formula.
**Formula for Expected Value (Mean) of a Hypergeometric Distribution:** The expected number of lands $E(X)$ in a 7-card hand, given a deck of $N$ cards with $K$ lands, is: $E(X) = \frac{7 \cdot K}{N}$. We want $E(X) = 3$. Given $N = 99$ cards, we can solve for $K$: $3 = \frac{7 \cdot K}{99}$. Solving for $K$: $K = \frac{3 \cdot 99}{7} \approx \frac{297}{7} \approx 42.43$. Since we can't include a fraction of a land, we round up to 43 lands.
**Check with a Smaller Example:** Let's verify with a smaller deck (e.g., 10 cards with 4 lands): Expected lands in a 3-card hand: $\frac{3 \cdot 4}{10} = 1.2$. Similarly, in a 7-card hand from 99 cards with 43 lands: $E(X) = \frac{7 \cdot 43}{99} \approx \frac{301}{99} \approx 3.04$, which is approximately 3 lands.
**Conclusion:** To have an average of 3 lands in your starting 7-card hand, include 43 lands in your 99-card deck. This ensures that the expected value aligns with your goal, though individual hands will vary due to randomness. |
| Paradigm 3 | To determine how many lands you should include in a 99-card Magic: The Gathering deck to have an average of 3 lands in your starting 7-card hand, we can use the concept of probability and expected value.
**The Basic Idea:** The probability of drawing exactly $k$ lands in a 7-card hand from a deck containing $n$ lands out of 99 total cards can be calculated using the hypergeometric distribution. The expected value (average) of the number of lands in the starting hand is given by: $E(X) = \frac{7 \times n}{99}$. We want $E(X) = 3$, so we can set up the equation: $\frac{7 \times n}{99} = 3$.
**Solving for $n$:** Let's solve for $n$: $7n = 3 \times 99$, $7n = 297$, $n = \frac{297}{7} \approx 42.4286$. Since the number of lands must be an integer, we round up to the nearest whole number.
**Result:** To have an average of 3 lands in your starting 7-card hand in a 99-card deck, you should include 43 lands. Verification: $\frac{7 \times 43}{99} = \frac{301}{99} \approx 3.0404$. This is close to 3, meeting the desired average.
**Why the Hypergeometric Distribution?** The hypergeometric distribution is appropriate here because: 1. We are drawing a fixed sample size (7 cards) without replacement from a finite population (99 cards). 2. There are two distinct groups: lands and non-lands. The formula for the probability of drawing $k$ lands is: $P(X = k) = \frac{\binom{n}{k}\binom{99-n}{7-k}}{\binom{99}{7}}$ where $\binom{a}{b}$ is the binomial coefficient (number of ways to choose $b$ items from $a$). The expected value $E(X)$ simplifies to $\frac{7n}{99}$, as mentioned earlier.
**Conclusion:** In a 99-card Magic deck, including 43 lands will give you an average of approximately 3 lands in your starting 7-card hand. Adjustments can be made based on your playstyle (e.g., card draw or mana ramp), but 43 is a solid starting point. |

Table 4: Case Study: Magic the Gathering Deck Land Calculation

