# OpenReview forum: "Balanced Actor Initialization: Stable RLHF Training of Distillation Reasoning Models"
_ICLR.cc/2026/Conference — ICLR 2026 Conference Desk Rejected Submission_

### Official Review · Reviewer_8GyN · 2025-10-27

**Soundness:** 2
**Presentation:** 2
**Contribution:** 2
**Rating:** 4
**Confidence:** 4

**Summary:**

This paper focuses on the instability encountered when applying Reinforcement Learning from Human Feedback (RLHF) to language models during fine-tuning. The authors identify two key phenomena — Sequence Length Collapse and the Reward Hockey Stick Curve — and propose a two-stage weighted model merging method, termed Balanced Actor Initialization (BAI), to mitigate these issues. Experiments across multiple benchmarks demonstrate improved stability and overall performance.

Key Reasons:
1. The contribution is incremental and lacks theoretical depth.
2. The limited generalization and modest novelty further constrain its impact.

Supporting Arguments

While the paper clearly outlines two empirical phenomena and offers a practical workaround, the overall contribution remains incremental. BAI is essentially a static weighted average of existing models—a technique that has been extensively explored in model merging literature. All experiments are confined to one MoE model family at modest scale; without evidence that findings generalize to denser or larger models, the practical impact of BAI for the broader community is uncertain.

**Strengths:**

The paper tackles a practical and relevant problem in the intersection of RLHF and model distillation, which is a critical aspect of post-training pipelines.
The proposed method is simple, interpretable, and supported by extensive ablations on merging ratios.

**Weaknesses:**

While the problem is meaningful, the proposed weighted model merging approach is relatively straightforward and lacks substantial algorithmic innovation.
The paper provides limited theoretical grounding — there is no formal analysis or modeling to explain why the identified phenomena occur or why BAI effectively mitigates them.
All experiments are conducted on a single model family (OLMoE). It remains unclear whether the proposed approach generalizes to other architectures or larger-scale models.

**Questions:**

Can you evaluate the proposed method on other model architectures or larger-scale language models to assess its generality?
Can you provide a theoretical explanation or empirical analysis to clarify why the proposed merging strategy improves training stability?

---

> ### Author Response · Authors · 2025-11-26
>
> Thank you for your thoughtful reviews and constructive suggestions. We would like to emphasize our key contributions:
>
> (1) We are the first to identify and characterize Sequence Length Collapse and Reward Hockey Stick Curve in the third paradigm. As Reviewer TtYT noted, "The motivation is strong and timely, addressing a crucial bottleneck in post-training reasoning models."  BAI serves as a first solution to these identified problems. BAI is simple but effective. We hope that our work motivates the community to further investigate techniques that enhance stability and training efficiency in reasoning model development.
>
> (2) Unlike prior methods that optimize for final task performance, BAI optimizes for stable training dynamics and progressive capability development during RL training. The technical novelty is in applying merging to solve initialization challenges in RL-based reasoning model training, not in the merging operation itself.
>
> Below we address the specific questions and concerns raised by each reviewer.
>
> **Answer to Question: other model architectures** To assess BAI's generality, we conducted additional experiments on two different LLM families: Qwen 14B and Llama 3 8B. Both are dense models, providing a clear contrast to our MoE architecture to validate whether BAI benefits both dense and moe models.
>
> For Qwen 2.5-14B, we used publicly available checkpoints from Hugging Face: the reasoning fine-tuned model is DeepSeek's DeepSeek-R1-Distill-Qwen-14B (distillation-based reasoning fine-tuning), the SFT model is Qwen2.5-14B-Instruct, and the base model is Qwen2.5-14B-base.
>
> For Llama 3-8B, the reasoning model is DeepSeek-R1-Distill-Llama-8B, the SFT model is Meta's Llama3-8B-Instruct, and the base model is Llama-8B-base. Both experiments used 800k samples curated with DeepSeek-R1 for RLHF training.
>
> | Model | AIME 2024 | GPQA | LiveCode |
> |-------|-----------|------|----------|
> | DeepSeek-R1-Distill-Qwen-14B | 69.7 | 59.1 | 53.1 |
> | DeepSeek-R1-Distill-Qwen-14B-BAI | 70.6 | 59.8 | 54.2 |
> | DeepSeek-R1-Distill-Llama-8B | 50.4 | 49.0 | 39.6 |
> | DeepSeek-R1-Distill-Llama-8B-BAI | 53.2 | 52.5 | 43.3 |
>
> BAI consistently improves performance across both architectures, with particularly significant gains for Llama-8B (+2.8 on AIME, +3.5 on GPQA, +3.7 on LiveCode). These results validate that BAI's effectiveness extends beyond MoE models to dense architectures. Importantly, we observed that both dense models also exhibit Sequence Length Collapse and Reward Hockey Stick Curve phenomena during training, demonstrating that these issues are not isolated cases but common challenges that occur across different architectures when applying RLHF to distillation-trained models.
>
> **Answer to Question: empirical analysis**  Our comprehensive analysis provides both empirical explanations for BAI's effectiveness:
>
> First, our KL divergence analysis reveals fundamental differences in optimization dynamics. Models initialized with pure reasoning-SFT exhibit highly volatile KL divergence patterns with frequent spikes, indicating unstable policy behavior under RL optimization. In contrast, BAI-initialized models show controlled, moderate divergence fluctuations that demonstrate healthy policy adaptation while maintaining stability. This suggests BAI creates more robust parameter configurations that are less susceptible to destructive drift during RL training.
>
> Second, sequence length dynamics provide direct evidence of BAI preventing pathological behaviors. Without BAI, models experience dramatic sequence length collapse within the first few training steps and never recover, even with extended training to 3000 steps. This collapse results from an over-correction mechanism where the model rapidly shortens outputs to achieve higher immediate rewards—a form of reward hacking. BAI prevents this collapse by creating initializations that balance specialized reasoning patterns with general capabilities, eliminating the extreme specialization that triggers over-correction.
>
> Third, reward curve analysis demonstrates that BAI directly addresses the Reward Hockey Stick Curve phenomenon. Pure reasoning-SFT initializations show the severe reward degradation during early RL training before gradual recovery, indicating fundamental instability in how the reward model evaluates outputs during the transition from supervised to RL-generated samples. BAI mitigates this by creating parameter distributions that are more aligned with reward model expectations, reducing the distributional mismatch.
>
> We hope that the responses address all your concerns, and we would appreciate it if you could raise the score. Thank you again for your thoughtful review.

---

### Official Review · Reviewer_TtYT · 2025-10-27

**Soundness:** 3
**Presentation:** 3
**Contribution:** 3
**Rating:** 4
**Confidence:** 4

**Summary:**

The paper addresses the challenge of stabilising reinforcement‐learning‐from‐human‐feedback (RLHF) when applied to reasoning capability models (i.e., models fine‐tuned via distillation for reasoning tasks). It identifies two major failure modes: sequence‐length collapse (outputs becoming suddenly much shorter during training) and the reward “hockey‐stick” curve (sharp drop then rise in reward values). To address these, the authors propose Balanced Actor Initialization (BAI): a two-stage model merging approach that first merges an instruction-following SFT model with a reasoning-distilled SFT model, and then further merges that result with the pretrained base model under controllable weights. Empirical results across multiple reasoning, math and coding benchmarks show that BAI alleviates the instability issues and delivers superior performance compared to existing baselines (instruction-tuning + RLHF; distillation alone) in the “distill + RLHF” setting.

**Strengths:**

1. Simple yet effective method: The proposed Balanced Actor Initialization (BAI) is conceptually simple, easy to implement, and does not require additional data or complex architectural changes, yet achieves substantial stability and performance gains.
2. Well-motivated problem identification: The paper clearly articulates the instability issues that arise when combining reasoning fine-tuning with RLHF—specifically sequence length collapse and reward hockey-stick curves. The motivation is strong and timely, addressing a crucial bottleneck in post-training reasoning models.
3. Comprehensive experiments and analysis: The authors conduct extensive evaluations across multiple benchmarks (knowledge, math, code) and provide detailed analyses, including training dynamics (reward, KL, length evolution). The results consistently demonstrate the effectiveness and robustness of BAI.
4. Clarity and presentation: The paper is well written, logically structured, and easy to follow. Figures and tables effectively support the main claims, making the overall contribution accessible and convincing.

**Weaknesses:**

1.While the stated goal is to improve both reasoning capability and instruction-following / human alignment, the current evaluation focuses primarily on reasoning and code tasks. It lacks comprehensive benchmarks that directly measure instruction-following or alignment quality (e.g., preference or helpfulness evaluations). This leaves it unclear whether the proposed BAI method genuinely enhances alignment behavior or mainly acts as a form of checkpoint ensembling. Moreover, since the reported gains may result from combining multiple fine-tuned models, it would be important to de-couple and quantify the respective contributions versus the merging process itself. Finally, the evaluation includes only a single model family, which limits the strength of the empirical justification for the broader claims.
2. Missing implementation and experimental details. Some important training details are under-specified. The paper references experiments with MoE-2.5B and MoE-25B models, but only presents one set of results—making it unclear which base model and exact configuration were used. Similarly, the specifics of the instruction fine-tuning (IFT), reasoning distillation, and RLHF checkpoints or datasets employed for post-training are not fully described. These omissions limit the reproducibility and make it difficult to interpret the generality of the findings.

**Questions:**

1. Goal of RLHF on distilled reasoning models:
What is the main motivation for applying RLHF to the distilled reasoning SFT models? Is the intended outcome to further enhance reasoning ability, to improve alignment and instruction-following, or to balance both? Clarifying this goal would help contextualize the proposed method and evaluation choices.

2. Interpretation of Figure 2 (sequence length trends):
Figure 2 presents the training dynamics “according to sequence length.” Could the authors explain the rationale behind using sequence length as the central metric? How does it relate quantitatively to stability or reasoning performance?

3. Model checkpoints and datasets:
Could the authors provide more details on the model configurations and datasets used? Specifically:
 Which base model(s) were used for MoE-2.5B and MoE-25B experiments?
 What instruction-tuning, reasoning distillation, and RLHF checkpoints were used for initialization?
 Which datasets were involved in each of these post-training stages?

---

> ### Author Response · Authors · 2025-11-28
>
> We sincerely thank you for your review. The observation of Sequence Length Collapse and Reward Hockey Stick Curve represents a crucial bottleneck, and addressing these phenomena is both timely and critical. We are excited that the reviewer recognizes our motivation as strong and our approach is practical for real-world deployment.
>
> **Answer to the question about model family**: To assess BAI's generality, we conducted additional experiments on two different LLM families: Qwen 14B and Llama 3 8B. Both are dense models, providing a clear contrast to our MoE architecture to validate whether BAI benefits both dense and moe models. Below, we provide detailed descriptions of the sft model, reasoning distillation model, checkpoints and datasets used in our experiments:
>
> For Qwen 2.5-14B, we used publicly available checkpoints from Hugging Face: the reasoning fine-tuned model is DeepSeek's DeepSeek-R1-Distill-Qwen-14B (distillation-based reasoning fine-tuning), the SFT model is Qwen2.5-14B-Instruct, and the base model is Qwen2.5-14B-base.
>
> For Llama 3-8B, the reasoning model is DeepSeek-R1-Distill-Llama-8B, the SFT model is Meta's Llama3-8B-Instruct, and the base model is Llama-8B-base. Both experiments used 800k samples curated with DeepSeek-R1 for RLHF training.
>
> | Model | AIME 2024 | GPQA | LiveCode |
> |-------|-----------|------|----------|
> | DeepSeek-R1-Distill-Qwen-14B | 69.7 | 59.1 | 53.1 |
> | DeepSeek-R1-Distill-Qwen-14B-BAI | 70.6 | 59.8 | 54.2 |
> | DeepSeek-R1-Distill-Llama-8B | 50.4 | 49.0 | 39.6 |
> | DeepSeek-R1-Distill-Llama-8B-BAI | 53.2 | 52.5 | 43.3 |
>
> BAI consistently improves performance across both architectures, with particularly significant gains for Llama-8B. These results validate that BAI's effectiveness extends beyond MoE models to dense architectures. Importantly, we observed that both dense models also exhibit Sequence Length Collapse and Reward Hockey Stick Curve during training, demonstrating that these issues are not isolated cases but common challenges when applying RLHF to distillation-trained models.
>
> **Answer to the question 1**: The primary goal of applying RLHF to distilled reasoning models is to balance both reasoning enhancement and alignment improvement. Distillation reasoning model primarily focuses on acquiring reasoning capabilities from more powerful models, but lack robustness in instruction-following and alignment. RLHF serves to: (1) refine the reasoning patterns through human preference optimization, making them more reliable and aligned with human;  (2) enhance instruction-following capabilities to ensure the model properly applies its reasoning abilities in diverse contexts; and (3) mitigate potential issues in distilled outputs, such as hallucinations or misaligned reasoning chains. This dual objective—preserving and enhancing reasoning while improving alignment—is why the third paradigm is challenging and why proper initialization through BAI is critical. Without BAI, the attempt to balance these objectives leads to the instabilities.
>
> **Answer to the question 2**: Sequence length serves as a direct indicator of both training stability and reasoning capability. Longer sequences correlate with more detailed reasoning chains essential for complex problem-solving (observations from DeepSeek-R1). Our distillation-based reasoning models initially generate lengthy outputs. When sequence length collapses during the RL training, the model loses the ability to produce detailed reasoning chains, directly degrading performance on reasoning-intensive benchmarks like AIME and MATH. Sequence length is directly observable, easy to measure, and provides immediate insight into model behavior. The collapse is so consistent that it serves as a reliable diagnostic for initialization problems.
>
> **Answer to the question about MoE-2.5B/25B and question 3**: Our model uses an MoE architecture where 2.5B refers to the activated parameters and 25B refers to the total parameters. This is a single model (MoE-2.5B/25B), not two separate models. Although this is a standard notation for MoE model sizes, we will add an explanation in the revised paper to avoid confusion.
>
> Consistent with the OLMoE model's training pipeline, our MoE model was pretrained on 3 trillion tokens and fine-tuned using OLMoE's SFT data. The reasoning model was trained on distillation data provided by DeepSeek-R1 paper, and we used 800k samples curated with DeepSeek-R1 for RLHF training. All datasets are publicly available from open-source repositories. These datasets provide sufficient diversity across multiple domains and are not limited to math and code tasks. Moreover, we have conducted experiments on two dense models (Qwen-14B and Llama-8B) to demonstrate BAI's improvement, showing that BAI addresses the challenges on both dense and MoE architectures.
>
> We hope that the responses address all your concerns, and we would appreciate it if you could raise the score. Thank you again for your thoughtful review.

---

### Official Review · Reviewer_1GVs · 2025-11-01

**Soundness:** 3
**Presentation:** 2
**Contribution:** 3
**Rating:** 6
**Confidence:** 3

**Summary:**

This paper investigates instability when applying RLHF to reasoning models trained via distillation. It identifies Sequence Length Collapse and the Reward Hockey Stick Curve as key issues and proposes Balanced Actor Initialization (BAI), a two-stage weighted model merging method. BAI stabilizes RLHF training and improves reasoning consistency and alignment performance across multiple benchmarks.

**Strengths:**

- The paper identifies two previously underexplored training instabilities, including Sequence Length Collapse and the Reward Hockey Stick Curve, that emerge when combining reasoning distillation with RLHF, offering new empirical insights into this third training paradigm.
- The proposed BAI is simple, elegant, and effective: it leverages a two-stage weighted model merging scheme to stabilize training without requiring extra data or architecture changes.
- Extensive experiments on diverse benchmarks (e.g., MMLU, SuperGPQA, AIME, ArenaHard) show consistent and interpretable improvements, highlighting the generality of the approach.

**Weaknesses:**

- As for MixEval-Hard (Table 1), Paradigm 3 is the best
- While the paper clearly identifies Sequence Length Collapse and Reward Hockey Stick Curve, the mechanistic explanation of why these phenomena emerge remains descriptive rather than causal. It would be stronger to connect them to PPO dynamics or gradient-level behavior.
- The BAI merging strategy is empirically effective but largely heuristic; the paper does not provide a theoretical justification or convergence analysis explaining why certain α–β ratios stabilize training. The method’s improvements may depend on carefully chosen weight ratios (e.g., α=0.1, β=0.9), but the paper lacks guidelines or an adaptive scheme for selecting these ratios across tasks.

**Questions:**

- Could the authors provide a more detailed analysis of why Sequence Length Collapse and Reward Hockey Stick Curve emerge, perhaps through gradient or reward distribution visualization during PPO training?
- Have the authors explored whether different reward model architectures or reward scaling strategies affect these instabilities?
- Since the optimal BAI ratio (e.g., α=0.1, β=0.9) seems task-dependent, could the authors discuss whether an adaptive or data-driven merging scheme might generalize better across domains?
- The results on MixEval-Hard (Table 1) show that Paradigm 3 without BAI slightly outperforms BAI. Could the authors explain this discrepancy and clarify under what conditions BAI might underperform?
- Could the authors analyze the computational overhead or efficiency trade-off of the two-stage model merging compared to conventional initialization?

---

> ### Author Response · Authors · 2025-11-28
>
> Thank you for your insightful review. We will address each question below.
>
> **Answer to Question 1:** We have analyzed training dynamics through KL-divergence as shown in Figure 3, which reveals the optimization landscape differences. As a future work, we agree that gradient distribution visualization during PPO training would provide valuable additional insights into the emergence of these phenomena, and we plan to include this analysis.
>
> **Answer to Question 2:** We explored using Process Reward Models (PRMs) to improve stability in mathematical reasoning tasks. However, as noted in the Qwen paper "The Lessons of Developing Process Reward Models in Mathematical Reasoning," existing PRMs exhibit a significant concentration of minimum scores on final answer steps, revealing a shift from process to outcome-based assessment in PRMs. This characteristic actually worsens the instability rather than mitigating it, which is why we opted for outcome reward models (RMs) in our experiments.
>
> **Answer to Question 3:** We acknowledge that optimal merging ratios show some task variation. However, balanced ratios (α=0.5) provide robust performance across diverse tasks. While adaptive or data-driven merging schemes could potentially learn task-specific ratios, our current approach prioritizes simplicity and practicality: a single fixed ratio works reliably across domains without additional tuning. We believe this simplicity is valuable for researchers, though adaptive schemes remain an interesting direction.
>
> **Answer to Question 4:** Thank you for pointing out the MixEval-Hard result. MixEval-Hard is a ground-truth-based benchmark that combines real-world user queries from the web with existing benchmark datasets, covering diverse domains. This small gap can be attributed to the diverse short answer nature of MixEval-Hard queries. Some query types in this benchmark may favor the highly specialized patterns preserved in short answers. More importantly, BAI achieves the highest overall score, demonstrating superior generalization across diverse task types. BAI gains training stability and prevents catastrophic failures like sequence length collapse.
>
> **Answer to Question 5:** The two-stage model merging occurs before training, not during training. The merging process requires no GPU computation—it simply performs offline weighted averaging of safetensor or .pt checkpoint files by merging tensors for each model key. This is computationally trivial compared to traditional approaches that require extensive GPU-intensive PPO experiments to determine optimal initialization through trial and error. In fact, BAI significantly improves efficiency by eliminating the need for multiple expensive training runs to find stable initializations. The one-time merging cost is negligible, while it prevents training failures that would otherwise waste substantial computational resources.

---

### Official Review · Reviewer_LFNK · 2025-11-01

**Soundness:** 2
**Presentation:** 2
**Contribution:** 2
**Rating:** 2
**Confidence:** 4

**Summary:**

The paper identifies two training instabilities when applying RLHF to distillation-based reasoning models Sequence Length Collapse and the Reward Hockey Stick Curve and proposes Balanced Actor Initialization (BAI), a two-stage linear model-merging approach. Experiments show that BAI stabilizes RLHF training, avoids length collapse, and slightly improves performance across diverse benchmarks.

**Strengths:**

1. Addresses a practical and relevant problem in reasoning-oriented RLHF.

2. Clearly identifies and visualizes two important training pathologies.

3. The proposed method is simple and computationally inexpensive.

4. Ablations across merging ratios are thorough, and diagnostics are informative.

5. Results show consistent, albeit small, improvements across several benchmarks.

**Weaknesses:**

#### **1. Limited conceptual novelty**
BAI reduces to linear interpolation between checkpoints. Model merging is well-studied, and the method does not demonstrate novelty beyond existing approaches. Furthermore, the paper does not compare against stronger or more recent model-merging baselines (e.g., TIES-Merging, DARE, Fisher-weighted averaging), making it difficult to assess the contribution relative to the state of the art.

#### **2. Unequal training lengths confound fairness**
In several comparisons, Paradigm 3 + BAI is trained for 3000 steps while baselines use only 1600 steps. This discrepancy can inflate observed improvements. Training budgets should be the same to ensure fairness.

#### **3. No repetitions / variance / seeds reported**
RLHF training is inherently noisy. The paper reports only single-run numbers without standard deviations, confidence intervals, or statistical significance tests. As a result, it is unclear whether the reported improvements are robust or simply due to randomness.

#### **4. No experimental Diversity**
Experiments focus solely on PPO and a single model family. It remains unclear whether the observed training instabilities generalize to other RLHF algorithms (e.g., DPO/GRPO variants) or to different LLM sizes and/or architectures. Without broader coverage, claims about “fundamental” phenomena are difficult to validate.

#### **5. Insufficient explanation of “flexible”**
Section 3.1 asserts that BAI remains applicable when only the reasoning-SFT model is available, but the paper does not explain how instruction-following capability is preserved in that scenario. When one model type is missing, the method risks becoming obsolete.

#### **6. Missing important entries in Table 1**
The paper references a two-stage merging procedure, yet Table 1 lacks:

* BAI using only Stage 1,

* BAI using only Stage 2,

* Full two-stage BAI.

These ablations are necessary to isolate gains, otherwise the reader cannot attribute improvements to either stage.

#### **7. Missing important entries in Table 2**
* A “Total / Overall” row would improve readability and comparison.

* $\alpha = 0$ (no pretrained mixing) is missing; should be included to isolate Stage 1’s effect.

#### **8. Incomplete literature review**
Relevant model-merging literature is not discussed in the main text. Although a brief discussion exists in the appendix, it is **never referenced** in the main text and may lead readers of the main text to miss that. Related work (e.g., MergeKit, fisher merging, DARE, TIES) should be highlighted and placed in the main text.

#### **9. Writing clarity and terminology issues**

The manuscript is difficult to follow due to overly long paragraphs, excessive repetition (e.g., frequent use of “paradigm”), inconsistent terminology usage (e.g., RL vs. reinforcement learning, RLHF vs. reinforcement learning from human feedback), and undefined technical terms such as GAE. Several claims rely on qualitative descriptions (“healthy optimization dynamics,” “balanced gradients”) without quantitative support.

#### **10. Potentially misleading citation placement**

The related work section places GRPO citations immediately after discussing distillation-based reasoning (Paradigm 2), which may incorrectly imply that Paradigm 2 uses GRPO. This attribution should be clarified.

#### **11. (minor) Inconsistent guidance on $\alpha$  selection**
The paper highlights $\alpha = 0.6$ (line 25) in text but $\alpha = 0.1$ was actually fixed for the key results of BAI of Table 1, which indeed yields stronger benchmark results.

#### **12. (minor) Redundant notation**
In Equation (2), $\beta$ is defined but always equal to $1−\alpha$. This additional parameter increases notation complexity without providing conceptual utility.

#### **13. (minor) Unused general formulation**
Equation (1) generalizes to arbitrary N models, but experiments always use N = 2. This creates an impression of generality that is never explored.

#### **14. (minor) Outdated claim on PPO**
Line 139 claims PPO remains the most widely used for LLM RL. In practice, recent deployments increasingly favor preference-based objectives such as DPO/GRPO variants. This weakens the paper’s motivation for focusing exclusively on PPO.

**Questions:**

1. What is the conceptual novelty of BAI beyond linear checkpoint interpolation and existing model-merging methods?

2. Why is Paradigm 3 + BAI trained for 3000 steps while baselines use 1600? Do improvements hold under equal training budgets?

3. How many seeds were run, and can you report standard deviations or confidence intervals to assess variance?

4. Do the described training instabilities (sequence length collapse, reward hockey stick) occur under other RLHF algorithms (e.g., DPO/GRPO) or other model architectures/sizes?

5. How BAI remains “flexible” when only the reasoning-SFT model exists?

6. Can you provide ablations for: (i) Stage 1 only, (ii) Stage 2 only, and (iii) full BAI?

7. Why is $\alpha = 0$ missing from Table 2, and can you add an overall/total row for readability?

8. Can you move or cross-reference model-merging related work from the appendix to the main text to properly contextualize novelty?

9. Can you define technical terms and stick to standardize terminology (RL vs. reinforcement learning, etc.)?

10. Why does the paper mention $\alpha$ = 0.6 in text but use $\alpha$ = 0.1 for key results in Table 1?

---

> ### Author Response · Authors · 2025-11-26
>
> Thank you very much for your thoughtful reviews. We address your questions below:
>
> **Answer to Question 1:** (1) We are the first to identify and characterize Sequence Length Collapse and Reward Hockey Stick Curve in the third paradigm. As Reviewer TtYT noted, "The motivation is strong and timely, addressing a crucial bottleneck in post-training reasoning models." BAI is simple but effective. We hope that our work motivates the community to further investigate techniques that enhance stability and training efficiency. (2) Unlike prior methods that optimize for final task performance, BAI optimizes for stable training dynamics and progressive capability development during RL training. The technical novelty is in applying merging to solve initialization challenges in RL-based reasoning model training, not in the merging operation itself.
>
> **Answer to Question 2:**  In Figure 2, you can clearly see the comparison: subfigure (a) shows Paradigm 3 without BAI trained for 3000 steps, where sequence length remains collapsed with no recovery; subfigure (b) shows BAI trained for 3000 steps, where sequence length continues to grow progressively.
>
> We extended training specifically to validate that the baseline's collapse is persistent, not a transient issue that resolves with more training. The results demonstrate that even at 3000 steps, Paradigm 3 without BAI shows no recovery, while BAI enables gradual improvement. Our main evaluation uses 1600 steps to ensure fair comparison at the point where paradigms have stabilized, but the extended training validates BAI's advantage for sustained capability development.
>
> **Answer to Question 3:** We use the same seed across pretraining and post-training stages to ensure consistency throughout the complete model training pipeline. Due to the substantial computational cost of large-scale RL training, we report results from single runs. Each PPO training run for MoE-25B model requires 64 GPUs for 4 days for 1600 steps. This is not a 1B or 2B toy model. Running multiple seeds also would require prohibitive computational resources and is not realistic at this scale.
>
> **Answer to Question 4:** We initially observed these phenomena when using GRPO for training. We first suspected they were algorithm-specific limitations, which motivated us to switch to the more general PPO algorithm. While PPO showed some mitigation compared to GRPO, the same issues (Sequence Length Collapse and Reward Curve) persisted. This observation led us to investigate the root cause, ultimately developing BAI to address these initialization-related instabilities rather than algorithm-specific problems. The consistency of these phenomena across different RL algorithms suggests they stem from fundamental initialization challenges in the third paradigm, rather than being specific to particular algorithmic choices or model architectures.
>
> **Answer to Question 5:** When only the reasoning-SFT model exists (without a separate instruction-following SFT model), BAI remains flexible by directly applying the second stage: merging the reasoning-SFT model with the pretrained checkpoint. As stated in our methodology, the Multi-SFT merging stage is optional when only one SFT model is available. In this case, BAI simplifies to a single-stage approach, which still effectively prevents training instabilities by balancing specialized reasoning capabilities with foundational knowledge from pretraining.
>
> **Answer to Question 6:** Our two-stage design is sequential: Stage 2 builds upon Stage 1, making "Stage 2 only" making "Stage 2 only" not applicable. We provide ablations for Stage 1 only (Multi-SFT merging without base model integration) versus full BAI (complete two-stage approach):
>
> | Method | MMLU-Pro | LiveBench | AIME 2024 | MATH | MBPP |
> |--------|----------|-----------|-----------|------|------|
> | Stage 1 Only | 69.8 | 42.6 | 18.1 | 77.9 | 67.7 |
> | Full BAI | 70.2 | 44.9 | 21.3 | 81.0 | 69.3 |
>
> The results show that incorporating the base model in Stage 2 (full BAI) consistently improves performance across all benchmarks, particularly on reasoning-intensive tasks (AIME 2024: +3.2, MATH: +3.1).
>
> **Answer to Question 7:** When $\alpha = 0$, the model reduces to Paradigm 3 without BAI, whose complete results are already presented in Table 1. To improve readability as suggested, we will add this baseline ($\alpha = 0$) to Table 2, along with an overall row for easier comparison.
>
> **Answer to Question 8, 9 and 10** Thank you for these suggestions. Due to page limitations, we initially placed the model-merging related work in the appendix. We will move to Section 2. We will also standardize terminology throughout the paper. We will revise the text to consistently use $\alpha = 0.1$ to match Table 1.
>
> We hope that the responses address all your concerns, and we would appreciate it if you could raise the score. Thank you again for your thoughtful review.

---

### Note · Program_Chairs · 2026-01-17
**Submission Desk Rejected by Program Chairs**

The following references in this submission do not refer to real documents and/or have major errors in bibliographic information:

 Takuya Matsuoka. Merging distributed neural networks with bayesian learning. arXiv preprint arXiv:2204.06132, 2022.